# Nanoparticle-Shielded dsRNA Delivery for Enhancing RNAi Efficiency in Cotton Spotted Bollworm *Earias vittella* (Lepidoptera: Nolidae)

**DOI:** 10.3390/ijms24119161

**Published:** 2023-05-23

**Authors:** Shelja Sandal, Satnam Singh, Gulshan Bansal, Ramandeep Kaur, Kanakachari Mogilicherla, Suneet Pandher, Amit Roy, Gurmeet Kaur, Pankaj Rathore, Anu Kalia

**Affiliations:** 1Regional Research Station, Punjab Agricultural University, Faridkot 151203, Punjab, India; shelja.1919@gmail.com (S.S.); rdeepraman23@gmail.com (R.K.); suneet@pau.edu (S.P.); sekhonmeet1990@gmail.com (G.K.); pankaj@pau.edu (P.R.); 2Pharmaceutical Sciences and Drug Research, Punjabi University, Patiala 140072, Punjab, India; gulshanbansal@rediffmail.com; 3Faculty of Forestry and Wood Sciences, Czech University of Life Sciences Prague, Kamýcká 129, 165 21 Praha, Czech Republic; mogilicherla@fld.czu.cz; 4Electron Microscopy and Nanoscience Laboratory, Punjab Agricultural University, Ludhiana 141004, Punjab, India; kaliaanu@pau.edu

**Keywords:** cotton, spotted bollworm, transcriptome, reference genes, dsRNA stability, nanoparticle encapsulated dsRNA

## Abstract

The spotted bollworm *Earias vittella* (Lepidoptera: Nolidae) is a polyphagous pest with enormous economic significance, primarily affecting cotton and okra. However, the lack of gene sequence information on this pest has a significant constraint on molecular investigations and the formulation of superior pest management strategies. An RNA-seq-based transcriptome study was conducted to alleviate such limitations, and de novo assembly was performed to obtain transcript sequences of this pest. Reference gene identification across *E. vittella* developmental stages and RNAi treatments were conducted using its sequence information, which resulted in identifying *transcription elongation factor* (*TEF*), *V-type proton ATPase* (*V-ATPase*), and *Glyceraldehyde -3-phosphate dehydrogenase* (*GAPDH*) as the most suitable reference genes for normalization in RT-qPCR-based gene expression studies. The present study also identified important developmental, RNAi pathway, and RNAi target genes and performed life-stage developmental expression analysis using RT-qPCR to select the optimal targets for RNAi. We found that naked dsRNA degradation in the *E. vittella* hemolymph is the primary reason for poor RNAi. A total of six genes including *Juvenile hormone methyl transferase* (*JHAMT*)*, Chitin synthase* (*CHS*)*, Aminopeptidase* (*AMN*)*, Cadherin* (*CAD*)*, Alpha-amylase* (*AMY*), and *V-type proton ATPase* (*V-ATPase*) were selected and knocked down significantly with three different nanoparticles encapsulated dsRNA conjugates, i.e., Chitosan-dsRNA, carbon quantum dots-dsRNA (CQD-dsRNA), and Lipofectamine-dsRNA conjugate. These results demonstrate that feeding nanoparticle-shielded dsRNA silences target genes and suggests that nanoparticle-based RNAi can efficiently manage this pest.

## 1. Introduction

*Earias* spp. (Lepidoptera: Nolidae), commonly known as the Asian spotted bollworm, is an important agricultural pest owing to economic losses in crops such as cotton and okra [1]. *Earias vittella* (Fabricius, 1794) is predominant in North Africa and the Indo Pakistan subcontinent [2]. Being cosmopolitan, the devastation caused by this pest is very severe, where an individual larva can destroy buds and bolls of the cotton crop. Although cultural, mechanical, biological, and chemical strategies control various pests, chemical pesticides generally remain the top preference of growers as they instantly knockdown the pest. On the other hand, the excessive use of chemical insecticides results in the persistence of these hazardous chemicals in the environment, causing several health hazards on non-target organisms including humans, and insecticide resistance in target pests [3]. Transgenic cotton has been a reliable and environmentally safe approach to control such harmful pests, but the prolonged exposure of insects to these toxins has resulted in the development of resistance and has jeopardized such technologies over time [4,5,6]. Considering the current scenario, there is a requirement for some novel and effective methods that should be economical, environment-friendly, and less likely to be overcome by pests [7,8]. As an alternative, RNA interference (RNAi) emerged as an effective strategy for controlling a pest by suppressing a particular gene of interest [9]. Previous studies have evidenced variable responses to RNAi across insect species belonging to the orders Lepidoptera, Hemiptera, and Diptera, such as the oriental armyworm (*Mythimna separata*) [10], leafhopper (*Nilaparvata lugens*) [11], cotton leafhopper (*Amrasca biguttula*) [12], peach–potato aphid (*Myzus persicae*) [13,14,15]. For instance, in the case of RNAi-sensitive coleopterans, it is highly efficient, and thus a minimal dose can result in effective and long-lasting RNAi, whereas in lepidopteran insects, it is less efficient even at much higher doses [15,16]. This variability among different insects suggests that some barriers influence RNAi efficiency in insects [17,18]. The dsRNA degradation in hemolymphs and reduced uptake of dsRNA by cells have been suggested as major reasons for the differential efficacy of RNAi among insects [15,16,19]. Therefore, a carrier system capable of entrapping, encapsulating, absorbing, or attaching active molecules for delivering dsRNA safely to the target site is considered pertinent. Recently, nanoparticles have received considerable attention as vectors for gene delivery. Nanoparticles are particulate dispersions or solid particles with particle sizes in the range of 10–1000 nm. Nanocarriers can protect RNAi molecules from enzymatic degradation and immune recognition and have much higher transportation efficiency across the cell membrane than other carriers [20,21]. Various nanoparticles already in use for gene delivery in different insects include silica nanoparticles (SNs), layered double hydroxides (LDHs) nanoparticles, liposomes, solid lipid nanoparticles (SLNs), and polymer nanoparticles [22,23,24].

Studies have demonstrated effective RNAi in *Anopheles gambiae* and *Spodoptera frugiperda* using chitosan–dsRNA nanoshields [23,25], *Aedes aegypti* and *Bemisia tabaci* with CQD-dsRNA nanoshields [22,26], and *S. frugiperda* and *Polistes dominula* with lipofectamine-dsRNA nanoshields [23,27] through feeding. Thus, we hypothesize that delivering nanoparticle-conjugated dsRNA in lepidopteran insects may improve RNAi efficiency. The success of RNAi also depends on the identification and screening of vital genes that can be knocked down in target insects [28]. To deploy RNAi, sequence information is a prerequisite; however, no genome or transcriptome data are available for *E. vittella* in public databases. Hence, this present study was taken up to provide sufficient insights into understanding gene expression in this pest. In the current study, a transcriptome was generated by sequencing the total RNA isolated from *E. vittella* fourth instar larvae, which was further utilized to find orthologs of genes encoding for proteins in the RNAi pathway and target genes for RNAi-mediated control of this insect. We also identified housekeeping genes for the RT-qPCR quantification of mRNA levels in the *E. vittella* life stages. Further, using chitosan, carbon quantum dots (CQD), and lipofectamine, we created dsRNA-nanoshields and successfully silenced the target genes, leading to pest mortality.

## 2. Results

### 2.1. De Novo Transcriptome

The 4th instar *E. vittella* RNA was used for a de novo transcriptome, generating 18,312,423 (2 × 75 bp) high-quality reads (Figure 1; Appendix A). The assembled reads yielded about 37,025 transcripts with an average length of 1050 bp per contig in *E. vittella* (Appendix A). TransDecoder predicted a total of 21,782 coding sequences (CDSs) with a minimum length of 297 bp, a maximum length of 9927, and a mean length of 981 (Appendix A). The BLASTx search against NCBI nr protein databases showed 95.8% (20,869) positive BLAST hits (Appendix A). The predominant positive BLAST hits were identified from insects such as *Amyelois transitella, Bombyx mori, Papilio xuthus, Papilio machaon, Danaus plexippus, Operophtera brumata, Papilio polytes, Plutella xylostella, Helicoverpa armigera, Manduca sexta, Spodoptera litura, S. frugiperda*, and *Spodoptera exigua* (Appendix A) with the highest similarity to *Amyelois transitella*. The gene ontology exposed 3358 CDSs under biological processes, 4047 under molecular, and 2636 under cellular components for GO categories (Appendix A). The major contribution towards the biological processes was from an organic substance metabolic process (19.5% and 16.97%), primary metabolic process (18.3 and 15.7%), and cellular metabolic process (15.98% and 14.1%) (Appendix A). Similarly, cellular components mainly consisted of 24.5% intracellular parts, 22.1% intrinsic components of membranes, and 20.5% intracellular parts, whereas molecular function primarily comprised ion binding (~18%), organic cyclic compound binding (~17%), and heterocyclic compound binding (~17%), respectively (Appendix A).

Only 5146 CDSs (23.6%) out of 21,782 were successfully annotated and were categorized into 23 different functional KEGG pathways (Appendix A). In addition, the functional annotation by KEGG analysis assigned the KO IDs to assembled transcripts under four main categories: metabolism, genetic information processing, environmental information processing, and cellular processes (Appendix A). Of these categories, the most significant contribution was toward metabolism (41.4%), followed by genetic information processing (23.1%). For pathways listed under these major processes, most CDSs were grouped into signal transduction (13.4%), transport and catabolism (11.3%), translation (8.6%), and carbohydrate metabolism (7%). A BUSCO analysis of the transcriptome assemblies revealed the presence of 77.9% complete BUSCO genes compared to the insect lineage (Appendix A).

### 2.2. Expression Profiles of Reference Genes

Initially, eleven candidate reference genes were selected and screened out based on their PCR amplification efficiency calculated with the help of a standard curve obtained using LightCycler software with a known concentration of a cDNA template (Appendix A). The amplified product analyzed on the agarose gel confirmed that all the genes are expressed in *E. vittella*, and further, the specificity of each gene was established by the presence of an expected amplicon size (Appendix A). The desired range for the amplification efficiency was 80–120%. The coefficient of determination (R^2^) for these reference genes ranged between 0.91 and 1.00 (Appendix A). The amplification specificity of each gene in the RT-qPCR analyzed by a melt curve analysis demonstrated the single peak. The Cq values for all the candidate reference genes were in the range of 14.33–37.51 (Figure 2).

### 2.3. Stability of Candidate Reference Gene Expression

Egg: Based on the overall ranking by NormFinder, ∆Ct, and RefFinder, *TEF* and *TATA* was designated as the highly stable genes in eggs (Table 1; Appendix A). *ACTIN*, *SOD*, and *ALP* were the least stable genes as calculated by all the algorithms (Table 1; Appendix A).

First instar: All algorithms calculated *TEF* and *SOD* to be consistently stable in the first instar, except Bestkeeper (Table 1; Appendix A). Based on Bestkeeper, *GADPH*, and *ACTIN* were the most stable genes, but the rest of the algorithms ranked these genes among the least five stable genes (Table 1).

Second instar: *TUBULIN* and *EFB* expression in this stage appeared to be consistently stable as per all the algorithms used for the analysis (Table 1; Appendix A). Only results based on the Bestkeeper analysis were moderately deviated, which ranked *EFB* among the five least stable genes.

Third instar: The *V-ATPase* and *ACTIN* were calculated as the most stable genes, whereas *APN* was ranked as the least stable gene in all the algorithms (Table 1; Appendix A).

Fourth instar: *TATA* and *ACTIN* were ranked as the most stable genes, whereas *ALP, APN*, and *SOD* were among the least stable genes in this instar as per all the algorithms (Table 1; Appendix A).

Fifth instar: *GADPH* and *SOD* were considered the most stable genes in *E. vittella* based on an overall ranking by RefFinder, whereas the Bestkeeper analysis placed *SOD* in sixth place and geNorm placed *GADPH* in seventh place in the ranking order (Table 1; Appendix A). *APN* and *V-ATPase* were ranked as the least stable genes in all the algorithms used in this study (Table 1).

Adult female: In the case of *E. vittella* adult females, *V-ATPase* and *GADPH* were concluded as the best genes with all the algorithms (Table 1; Appendix A). *ALP* and *TATA* were considered the least stable genes as per the calculations of all the algorithms (Table 1).

Adult male: The comprehensive ranking based on RefFinder as well as all the other algorithms designated *TEF* and *ALP* as the most stable genes; however, the results of Bestkeeper slightly deviated from all the algorithms, placing *TEF* and *ALP* at fourth and sixth position in the stability ranking, respectively (Table 1; Appendix A). All the algorithms designated *ACTIN* as the least stable gene in *E. vittella* adult males (Table 1).

All stages: The *RPS15* and *GAPDH* were calculated as the most stable genes among all the stages, whereas all algorithms designated *SOD* as the least stable across all the stages of *E. vittella* (Table 1; Appendix A).

### 2.4. Identification of RNAi Machinery Genes

Using the transcriptome data, we identified RNAi pathway genes, including the RNAi core machinery, dsRNA uptake, intracellular transport, nucleases, and RISC factor genes, to comprehend the RNAi mechanism in this insect (Figure 3). We observed that the core RNAi machinery genes *viz., Arganoute 2*, *Dicer 2*, and *Piwi* are significantly expressed in all life stages of *E. vittella*. *Transferrin, CHC isoform X2*, *Innexin*, *SORL-like*, and *SORL* genes are substantially expressed during dsRNA uptake. *V-ATPase B*, *V-ATPase G*, *V-ATPase C*, *V-ATPase F*, *V-ATPase D isoform X1*, *ADP-ribosylation factor*, *TERF1*, and *Rab7* are highly expressed in intracellular transport. While *RNA helicase isoform X2*, *DHX8*, and *DHX30 isoform X1* were substantially expressed in RISC factors, the *UTP14-like* isoform is highly expressed in nuclease (Figure 3).

### 2.5. dsRNA Degradation Studies

The crude hemolymph of *E. vittella* comprised 1.4 g/mL of total protein, which was subsequently diluted to produce concentrations of 0.97 g/mL, 0.12 g/mL, 0.015 g/mL, and 0.002 g/mL. A gel retardation assay was performed to confirm the degradation of dsRNA after its incubation with hemolymph (Figure 4). Furthermore, it was found that both the hemolymph concentration and the incubation time had a direct impact on the degree of dsRNA degradation. The higher dilutions of hemolymph retained the integrity of dsRNA, signifying the key role of dsRNA degrading nucleases in hampering RNAi efficiency (Figure 4). The dsRNA degradation was also directly proportional to the time of exposure, as the extended incubation time (5 h) increased the dsRNA exposure to dsRNases, leading to its complete degradation (Figure 4).

### 2.6. Identification and Expression Analysis of Functional Genes across Developmental Stages

In the transcriptome data, we identified functional genes such as *cytochrome P450*, *glutathione S transferase*, *acyl co-A*, *multidrug resistance mutation A*, *esterase*, *chymotrypsin*, *odorant binding protein*, *juvenile hormone*, *vitellogenin*, *cadherin*, *alpha-amylase*, *chitin synthase*, *V-ATPase*, and *aminopeptidase* (Figure 5). The RNAi machinery genes such as *RNA binding protein*, *Piwi*, *Dicer*, and *dsRNase* have also been identified in *E. vittella* from the transcriptomic data (Figure 5). The identified putative transcripts were then screened for the unique and common transcripts from the de novo transcriptome assembly, followed by removing duplicate sequences of respective genes using multiple sequence alignment. Sequence-based phylogenetic analysis by constructing maximum-likelihood trees suggested the evolutionary relationship of *cadherin* (Appendix A), *cathepsin* (Appendix A), and xenobiotic degradation CYP450 family (Appendix A) from *E. vittella* with their homologs from other organisms. KEGG-based annotation was performed to understand the biological functions of these transcripts, which revealed the different families for each gene. The RT-qPCR analysis showed an altered expression of different functional genes across the two stages of morphogenesis, i.e., larvae to pupae and from pupae to adults (Figure 5).

### 2.7. Nanoparticles’ Characterization and dsRNA Conjugation

TEM observation of the prepared naked chitosan and CQDs revealed that both the nanoparticles exhibited semi-spherical morphologies. The particle size dimensions were larger due to aggregate formation ranging from 30 to 50 and 50 to 80 nm for chitosan and CQDs, respectively. Whereas, the chitosan-dsRNA and CQD-dsRNA nanoconjugates showed a decrease in size ranges from 20 to 30 and 15 to 30 nm, respectively. This suggests that the synthesized dsRNA-nanoconjugates have stabilized and will aid in protecting and efficiently delivering dsRNA into the cells. (Appendix A). Furthermore, the gel retardation assay confirmed the binding of the dsRNA nanoconjugates efficiently (Appendix A).

### 2.8. RNAi Efficiency

Feeding bioassays were set up to confirm the efficiency of nanoparticles to induce RNAi in *E. vittella.* Our earlier studies confirmed the uptake of dsRNA and dsRNA nanoconjugates by the lepidopteran cells and tissues [15,23]. However, the uptake of dsRNA nanoconjugates through feeding was confirmed through the oral administration of red food dye incorporated, which was visualized in the dissected gut and fecal matter of the insect (Appendix A). The results showed that efficient gene silencing is induced by dsRNA nanoconjugate feeding compared to naked dsRNA without causing much impact on the survival percentage of treated insects (Appendix A). This further overruled any impact of nanoparticles alone on larval survival. Feeding of the larvae with dsCadherin–Chitosan and dsCadherin–CQD conjugate significantly reduced the expression of the corresponding gene by 78.6% and 93.6% when compared to naked dsRNA-fed larvae (Figure 6). Additionally, feeding the dsAminopeptidase–Chitosan, dsAminopeptidase–CQD, and dsAminopeptidase–lipofectamine complex was statistically significant in causing the reduction in the gene expression level to 83.9%, 83.8%, and 81.3% respectively, in comparison to naked dsRNA (Figure 6). The results further suggested that the RNAi efficiency of dsAlpha-amylase–lipofectamine conjugate reduced its mRNA level by 84.4% (Figure 6), while a 70.6% and 76.6% reduction in mRNA level of the target gene has been recorded with dsAlpha-amylase–CQD and dsAlpha-amylase–Chitosan conjugates, respectively (Figure 6). Compared to feeding naked dsRNA, feeding dsChitin synthase, dsV-ATPase, and dsJuvenile hormone coupled with chitosan, lipofectamine, and CQD significantly reduced the level of gene expression (Figure 6). Overall, it has been concluded that feeding naked dsRNA to an *E. vittella* larva did not trigger efficient RNAi of the target genes. The dsRNA of various genes fed to *E. vittella* did not cause any significant mortality or result in any phenotype (Appendix A) except for dsChitin synthase, which resulted in a significant reduction in the percent pupation and adult emergence when a higher concentration of dsRNA, i.e., 20 µg, was fed twice with a gap of 72 h (Figure 7). However, no statistically significant difference in the percent pupation and adult emergence was observed in naked dsChitin synthase and dsGFP. The larval deformities and adults without hairy tuft, and the significant difference in percent pupation (36% less) and adult emergence (52% less) were observed in dsChitin synthase + Chitosan as compared to the dsGFP control (Figure 7).

### 2.9. Expression of Core RNAi Genes after Gene Knockdown

The triggering of the RNAi activity in *E. vittella* in response to the dsRNA exposure suggested a maximum expression response of core RNAi components (*RNA-binding protein*, *dsRNase*, *Dicer*, and *Piwi*) in the insects fed with chitosan-coated dsGFP compared to those fed with non-coated dsGFP. In addition, the expression of these genes was minimal in the control insects (Figure 8). The results imply that the RNAi machinery components express as soon as chitosan-coated dsRNA enters the organism, and hence coating could improve RNAi in the target insect (Figure 8).

## 3. Discussion

### 3.1. E. vittella Transcriptome Generation

The *E. vittella* is an oligophagous pest that affects several significant crops and reduces annual crop yields by millions of dollars [2]. However, this pest′s genetic resources are limited due to a lack of genomic and transcriptomic data. In this study, we sequenced the transcriptome of the *E. vittella* 4th instar larvae intending to identify target genes for RNAi while simultaneously generating a useful genetic resource (Figure 1; Appendix A; Appendix A). We embarked on a comprehensive study to identify useful target genes for RNAi and housekeeping genes for RT-qPCR because earlier successful RNAi in insects inspired us to do so. A total of 98% of the arthropod gene set was represented, according to the analysis of the *E. vittella* larval transcriptome (Appendix A).

### 3.2. Reference Genes Analysis

The normalization of the expression data is the prerequisite for any type of gene expression study with animal or plant species using traditional semi-quantitative or new-generation RT-qPCR. This requires a suitable housekeeping or reference gene whose expression is stable across the life stages or under various environmental stresses. The eleven reference genes for this insect were evaluated using four separate programs, namely geNorm, Normfinder, Bestkeeper, and delta CT (Appendix A). Our experimental data showed fluctuation in the expression of putative reference genes across various developmental stages, and this may be due to the high degree of stage-specific regulation in the transcriptional activity (Figure 2; Appendix A), thus emphasizing the significance of recognizing various stable genes specific to developmental stages. Earlier studies also suggested the difficulty of identifying a gene in an organism that is stable over developmental stages as well as varying experimental conditions [29]. So, our study successfully evaluated and validated different reference genes by RefFinder and identified them in the generated resource data in this insect (Table 1).

Recent studies have listed several stable reference genes under different conditions in different pests, including *Helicoverpa armigera* [30], *Amrasca biguttula* [12], *Bemisia tabaci* [31], *Leafminer liriomyza trifoli* [32], *Aphis gossypii* [33], *Phenacoccus solenopsis* [9,34], *Aedes aegypti* [35], *Lipaphis erysimi* [36], *Halyomorpha halys* [28], *Callosobruchus maculatus* [37], and *Ips typographus* [38]. The present studies reported *GADPH* among the top five reference genes in the fourth and fifth instar, adult males and females (Table 1). *GADPH* plays a vital role in forming membranes, microtubules, phosphotransferase, and kinase reactions [39]. The expression of this gene was also documented to be highly stable in the various development stages of *S. litura* larvae under temperature stress [40] as well as in the bacteria-infected bee *Apis mellifera* [41]. Among developmental stages, *EFB* was also among the most stable genes, such as *GADPH*, and these findings are consistent with previous studies in other insects where the expression of *EFB* has been used for the normalization of qPCR data [30,33,42]. *ACTIN* is involved in various essential cellular functions such as cell secretion, motility, maintenance of the cytoskeleton, and muscle contraction [43,44]. Its expression is highly variable over the developmental stages, thus making it unsuitable as a reference gene under some situations, as reported in earlier studies, such as various tissues of *B. mori* [45] and *S. exigua* [46]. In response to various stresses during their lifetime, insects tend to induce excessive reactive oxygen species (ROS) and to combat ROS, these creatures have evolved a complex network of enzymatic antioxidant systems. *SODs* are among the major components of this system, which convert superoxide into O_2_ and H_2_O_2_ [47]. In our study, the expression level of *SOD* was found to be most stable across the developmental stages of *E. vittella*. Previous studies also identified *SOD* as a stable gene in the developmental stages of *S. exigua* and *Tuta absoluta* [46,48]. *SOD* was also identified as the most stable reference gene under diverse experimental conditions such as starvation stress in *Thrips tabaci* [49], and temperature treatment in *Harmonia axyridis* [50], however in *C. elegans*, the expression of *SOD* was found to be variable [51]. The current studies with *E. vittella* designate it as highly stable and, thus, a suitable reference gene in some developmental stages.

### 3.3. Development and Detoxification Genes Expression Analysis

The higher expression of diverse detoxification genes in pre-pupal stages can be correlated to the active foraging behavior that allows them to defend themselves from toxic compounds produced by host plants [52]. In general, the insecticide resistance mechanisms in insects are commonly associated with one or more detoxification genes, comprising *P450s*, *esterases*, *mitogen-activated protein kinase*, and *glutathione S-transferases*. The insecticide resistance mechanism mainly involves the up-regulation and changes in the catalytic properties of enzymes involved in detoxification. *Glutathione S-transferase* (GST) and *esterases* are multifunctional enzymes that play a major role in insecticide detoxification either by direct metabolism or sequestration [53,54]. The *cytochrome P450 monooxygenases (P450s)* belong to a complex gene superfamily of heme-thiolate proteins involved in the growth and development, biosynthesis of hormones, and degradation of xenobiotic compounds [55,56,57]. The identification of several transcripts for *CYP450*, *cathepsin*, and *cadherin* genes from the transcriptome data and their phylogenetic analysis provided the evolutionary relationship of this insect with its respective homologs (Figure 5). Most of the transcripts in the insect belonged to the CYP3 clan, while the others were classified into CYP2, CYP4, and mitochondrial clans. A high number of transcripts for cathepsin belong to the digestive cysteine protease type, which is further divided into cathepsin B and cathepsin L subgroups. However, a few *cysteine proteases* were distantly related to the other *cysteine proteases*. In the case of *cadherins*, most of the transcripts were found to fall in the category of neural cadherin, which shows a minimal divergence in the species. We also identified the gene encoding enzymes of the sesquiterpenoid *juvenile hormone (JH)* and ecdysteroid biosynthetic pathways. Since *juvenile hormones* play a key role in physiological processes such as metamorphosis and insect reproduction [58,59,60], these could be considered potential targets for RNAi.

### 3.4. Identification and Expression Analysis of RNAi Machinery Genes

Three different RNAi pathways, siRNA, miRNA, and piwiRNA (piRNA), have been found in insects. The siRNA machinery is activated when dsRNA is successfully delivered to the cell. The enzyme *Dicer-2 (Dcr-2)* converts the dsRNA into siRNAs, which are then incorporated into the RNA-induced silencing complex (RISC). The remaining antisense strand then directs the RISC to the complementary mRNA strand in a sequence-specific way after *Argonaute2 (Ago2)* cleaves and eliminates the sense strand. The *AGO2* then breaks down the mRNA strand, which results in post-transcriptional gene silence [61]. Except for *E. vittella*, the discovery of the key genes for the RNAi machinery has been explicitly evaluated for several insects [9,28,62,63,64]. In *E. vittella*, for the first time, we looked at and verified the expression of RNAi core machinery and other related genes in the obtained transcriptome data. We found that the RNAi machinery genes in *E. vittella* exhibited a wide range of expressions (Figure 3). Earlier studies suggested that RNAi machinery is incredibly efficient and systemic in the insects belonging to Coleoptera insects because of the exclusive presence of *Staufen C* homologs that are a crucial role in dsRNA absorption and intracellular transport [65]. Our results are consistent with other studies that reported low *staufen* gene expression in *E. vittella*, similar to other lepidopteran insects (Figure 3). *Utp14* is required to recruit the RNA exosome effectively [66], and the expression of *UTP14-like* genes was high in *E. vittella*. Lepidopteran (*H. virescens*) insects degrade the injected or fed dsRNA more quickly than coleopteran (*L. decemlineata*) insects [16]. These studies also suggest that both the lepidopteran and coleopteran cell lines efficiently absorbed dsRNA and tissues; however, coleopterans converted the dsRNA to siRNA while lepidopterans did not process it into siRNA. Our hemolymph assay also showed a rapid degradation of dsRNA, possibly due to the *E. vittella* expressed dsRNA-specific nucleases. Based on past studies and our findings, we assumed that the dsRNA is efficiently taken by *E. vittella* cells and tissues but may be transported to endosomes and broken down in lysosomes.

### 3.5. dsRNA Degradation: A Potential Cause of Low RNAi Sensitivity

The responses of the insects from different orders and different insects within the order to RNAi vary, ranging from a better efficiency (coleopterans) to lesser efficiency (lepidopterans) [8]. The dsRNA, when fed or injected into the insect, encounters the nucleases, specifically the dsRNases, which may be considered the first point of a barrier for efficient RNAi. Thus, a poor RNAi response in lepidopterans may be attributed to the dsRNases specific gut nucleases, which degrade the dsRNA before it can go through the cellular process. The gel retardation assay in the present study demonstrated the degradation of dsRNA indicating that the degree of degradation was closely correlated with the content of hemolymph as well as the incubation time (Figure 4). These results showed that the crude hemolymph might be rich in dsRNases, causing the complete degradation of dsRNA followed by a proportional response of dsRNA degradation to hemolymph dilutions (Figure 4). Thus, this indicates that the dsRNases play a crucial role in RNAi in insects, specifically, the ones belonging to Lepidoptera. The exposure time also plays a critical role as it seems to be directly proportional to the dsRNA degradation. The extended incubation of 10 h led to the complete degradation of the dsRNA. However, in vivo, the exposure time has little relevance as the ingested or injected dsRNA may not possibly face such long exposures in the gut or hemolymph. Because the gut or hemolymph contains dsRNA-specific nucleases, numerous earlier investigations have also suggested these as one of the possible causes for dsRNA degradation in different insect species [9,15,16,26,67,68,69]. Lepidopteran insects′ stomach nucleases may prevent sufficient amounts and high concentrations of dsRNA from reaching the cells, which results in ineffective RNAi; however, employing nanoparticles increased the RNAi effectiveness [70,71,72]. Current studies have effectively shown dsRNA degradation in the hemolymph of *E. vittella* and that nanoparticles may aid in nuclease escape and enhance RNAi in this organism.

### 3.6. Feeding of Nanoshield-dsRNA Improves the RNAi in E. vittella

RNAi-mediated knockdown of different genes in many lepidopteran insects has been well demonstrated through feeding or injection [7]. However, the degradation of dsRNA by specific nucleases limits the process′s sensitivity in living organisms. The primary factor influencing the stability of dsRNA in insects is the nuclease enzymes present in the gut, which act as a first point of hurdle for efficient RNAi in lepidopteran insects. Many previous studies as well as our current study with *E. vittella* demonstrated that crude hemolymph/gut juices or a higher concentration of hemolymph/gut juices causes more degradation of dsRNA in comparison to controls and lower concentrations of hemolymph. This means that the dilution of hemolymph/gut juices is inversely proportional to the level of degradation. Complexation of a nanocarrier with dsRNA enhances the stability of the same by encapsulation, tethering and thereby shielding it from a nuclease attack in the gut or other terms, reducing the exposure of dsRNA to dsRNA-specific nucleases. Lepidopteran insect cells may endocytose dsRNA by clathrin-mediated endocytosis or SID-1 receptors [16]. The endosomal trapping of the dsRNA, once it enters the lepidopteran cells, leads to the escape of dsRNA from being exposed to RNAi machinery, and thus is not processed into siRNA. The nanoparticles may be playing a key role in bypassing this endosomal pathway, thereby improving the RNAi efficiency in these insect species. Thus, the slow cellular process of clathrin endocytosis for the delivery of dsRNA can be enhanced using nanoparticles [17,18]. Previous research has shown that nanoparticles serve as molecular carriers for dsRNA distribution, improving dsRNA persistence and cellular uptake [22,26]. Core–shell nanoparticles, BAPCs (branched amphiphilic peptide capsules), quantum dots, liposomes, guanylated polymers, and chitosan have all been employed in different insects to increase the effect of RNAi [30,73,74,75]. The nanoparticles, through electrostatic contact, bind their amino group with the phosphate group of the target dsRNA to establish an association between the two molecules [76]. Compared to synthetic compounds, chitosan is an inexpensive, natural, and biodegradable polymer [23]. The use of chitosan nanoparticles in present studies may have improved the intracellular transport of dsRNA, thus protecting it from dsRNase digestion and lowering the accumulation of dsRNA in the endosomes. Based on earlier studies and our present experimentation, the chitosan nano-dsRNA conjugate allowed the safe passage of dsRNA to the cytoplasm of *E. vittella* tissues, which was processed by dicer enzymes into siRNAs and enhanced the knockdown of target genes compared to naked dsRNA. Previous studies have demonstrated that chitosan–dsRNA nanocomplexes improved the RNAi efficiency in lepidopteran insects and mosquitoes [23,25]. The chitosan–dsRNA nanocomplexes were successfully employed for enhancing RNAi in mosquito species through the silencing of various target genes such as *chitin synthase* in *Anopheles gambiae* [25], the *sm* (*single-minded*) gene controlling olfactory system in both the larval and pupal stages of *Aedes aegypti* [77], the knockdown of the *cadherin* gene in *A. gambiae* to elucidate its role in *Cry11Ba* toxicity [78], and the knockdown of a *vestigial* gene associated with wing development that resulted in mortality, delayed development, and adult wing deformity in *A. aegypti* [79]. It has also been shown that chitosan-enabled efficient dsRNA delivery into insect cells can decrease shrimp viral infection in Sf9 cells by targeting the RdRp gene specific to the yellow-head virus [80]. The effectiveness of encapsulation was enhanced by cross-linking chitosan to sodium tripolyphosphate, which in turn imparted protection to dsRNA from nucleases resulting in better cellular uptake, dsRNA biodistribution, larval mortality, and efficiency of the gene in *A. aegypti* [75]. Compared to the performance of naked dsIAP, the improved knockdown efficiency with a dsIAP–chitosan conjugate and 20% mortality were observed in *S. frugiperda* [23]. In the present studies, we also observed that the chitosan-shielded dsRNA enhanced the RNAi by imparting protection from nucleases present in the lumen contents and hemolymph and thus increasing the gene knockdown in the lepidopteran pest *E. vittella*. Further, the dsChitin synthase coupled with chitosan showed a significant reduction in the pupation and adult emergence of *E. vittella*. The RNAi in lepidopterans with naked dsRNA has been a debatable issue and many studies have documented a poor RNAi response in various insects belonging to this order. However, nanoparticle intervention coupled with higher concentrations of dsRNA might be contributing to overcoming the dsRNases’ degradation or endosomal escape in lepidopteran insects [70,71,72,75]. In our studies, we also observed that feeding 10µg of naked or nano-shielded dsRNA of various genes failed to cause mortality or generate different phenotypes in *E. vittella*, however, increasing dsRNA to 40µg for dsChitin synthase resulted in a reduction in pupation and adult emergence as well caused low frequencies of larval and pupal deformities. Earlier studies with *S. frugiperda* also suggested that chitosan-coated dsRNA resulted in a decrease in pupal and adult weight compared to naked dsIAP [71].

Lipofectamine, a transfection reagent, has been used as a dsRNA carrier for efficient delivery and for enhancing RNAi efficiency in insect species. Cationic lipids bind nucleic acids to create transfection complexes, which are unilamellar liposome structures with a positive surface charge. The transfection complex is localized to the cell surface for endocytosis and transfection via charged interactions between the cationic lipid head group and the negatively charged cell membrane.

Injecting the dsRNA–lipofectamine 2000 complex into transgenic strains of *D. melanogaster* larvae carrying the widely expressed *β-glucuronidase* gene showed a 52% reduction in the target gene expression in treated animals [81]. The UGA-CiE1 cells derived from a *Chrysodeixis* (*Pseudoplusia*) transfected dsRNA with lipofectamine 2000 efficiently assimilated the complexes [82]. When the *rps13*, *COP*, and *vha26* genes were targeted by dsRNA-lipofectamine 2000 it showed significant mortality in spotted-wing *Drosophila suzukii* [83]. When evaluated in tick cell lines, nymphs, and adults, the siRNA and dsRNA encapsulated with lipofectamine 2000 demonstrated improved gene silencing [84,85]. In addition, we observed that the lipofectamine-shielded dsRNA improved the RNAi in present studies by reducing the impact of dsRNases present in the lumen contents and hemolymph of *E. vittella*.

The conjugation of dsRNA and CQD particles may be brought on by polyethyleneimine (PEI), a cationic polymer having a nucleic-acid-binding affinity. Faster transfection efficiency, increased stability, and buffering caused by this polymer complex result in an endosome break due to the osmotic pressure, which aids in internalizing dsRNA into the gut cell′s cytoplasm [86]. To effectively transport dsRNA-targeted functional genes to cause a gene knockdown against *C. suppressalis*, *A. aegypti*, and *B. tabaci*, CQD nanoparticles have been utilized [22,26,87]. In our investigation, the coupling of dsRNA with CQD improved the knockdown efficacy of the target genes by preventing dsRNA from being degraded by nucleases.

The efficient knockdown of any target gene may not always result in mortality; thus this may depend on the uniqueness of the trait governed by the single vital gene. The studies suggest that the knockdown of *Chitin synthase* shielded by chitosan-coated particles exhibited some phenotypic changes as well as a significant reduction in the survival percentage, adult emergence, and pupation in comparison to other nanoconjugates and targets selected for a knockdown. So, these studies need further investigations by selecting more targets that can harness the phenotypic effect in *E. vittella.*

## 4. Material and Methods

### 4.1. Rearing of Insect

Larvae of *E. vittella* collected from cotton fields of GPS coordinates (30°40′41.4696″ N, 74°44′22.3980″ E) in Faridkot, Punjab were reared on an artificial diet. The adult moths were fed with a 10% sucrose solution with the help of cotton swabs [88]. The insects were kept in the incubator at 65–70% relative humidity, 14:10 h light, dark photoperiod, and 27 ± 2 °C temperature.

### 4.2. Total RNA Isolation and Sequencing

Total RNA was isolated from the insect sample in TRIzol using Direct-Zol R.N.A. Miniprep Kit (ZYMO Research). The quality and quantity of the isolated RNA were analyzed using a 1.2% agarose gel and NanoDrop^TM^ 1000 (Thermo Fisher Scientific, Wilmington, DE, USA) reading, respectively. Paired-end RNA-Seq libraries were prepared using an Illumina TruSeq stranded mRNA library preparation kit. The fragments of Poly A-tailed mRNA were pulled out from the total RNA using poly-T attached magnetic beads and subjected to enzymatic fragmentation. Following the synthesis of the first strand of cDNA, the RNA-dependent synthesis of the second strand was facilitated using a strand and ACTIN-D mix. The double-stranded cDNA was then purified with XP (Ampure) beads, followed by adapter ligation, A-tailing, and a limited number of PCR cycles to enrich it. The PCR-enriched libraries were analyzed for quality and quantity in a 4200 TapeStation system (Agilent Technologies, Santa Clara, CA, USA) using High sensitivity D1000 Screen tape. The paired-end (PE) libraries were loaded onto NextSeq 500 (Illumina, San Diego, CA, USA) for cluster generation and sequencing (2 × 75 bp).

### 4.3. De Novo Transcriptome Assembly

The sequenced raw data were analyzed using a Trimmomatic v0.35 [89] by removing adapter sequences, ambiguous reads (reads with unknown nucleotides “N” larger than 5%), low-quality sequences (reads with more than a 10% quality threshold (QV) < 20 phred scores), and a minimum length of 75 bp after trimming was applied to obtain high quality reads, and these reads were used for the de novo assembly of both the samples. Various parameters such as Adapter trimming, Sliding window, Leading, and Trailing parameters were considered for filtering the reads below a threshold quality of 20. Velvet v1.2.10 [90] and Oases v0.2.09 [91] assembled the filtered high-quality reads into transcripts on optimized Kmer 23 and 27, respectively. For quantitative evaluation, Burrows–Wheeler Aligner BWA v0.7.12 was used to validate transcripts by mapping back the reads to their respective assembled transcripts. Further, TransDecoder (https://github.com/TransDecoder/TransDecoder/wiki, accessed on 20 August 2022) was used to predict the coding sequences (CDS) from transcripts, which were then searched using BLASTx against NCBI nr (non-redundant) protein databases. Blast2GO was used to determine the Gene ontology (GO) annotations of the CDSs. The predicted CDS functions were categorized using GO assignments. Each BLASTx functionally annotated CDS underwent GO mapping to retrieve GO keywords. The involvement of the predicted CDS of *E. vittella* in the biological pathway was determined by mapping the CDS about canonical pathways in KEGG [92]. The output of the KEGG analysis included KEGG Orthology (KO) assignments and the corresponding Enzyme Commission (EC) numbers and metabolic pathways of predicted CDSs using the KEGG Automatic Annotation Server (KAAS). To assess the completeness of the transcriptome, the Benchmarking Universal Single-Copy Orthologs (BUSCO v3.0.2) tool was used [93], and the analysis was performed against 1658 BUSCO sets, using 42 species of insect BUSCO lineage as a reference.

### 4.4. Selection, Amplification, and Real-Time Quantitative PCR (RT-qPCR) of Reference Genes

Preliminary studies were performed to identify the best reference gene among various genes selected based on previous studies on other insects and commonly used genes in lepidopteran insects. Eleven genes were selected from transcriptomic data of *E. vittella*, including *ACTIN (Actin-4)*, *TEF (Transcription elongation factor S-II)*, *RPS18 (28S ribosomal protein S18b)*, *V-ATPase (V-type proton ATPase)*, *TATA (TATA box binding protein)*, *EFB (Elongation factor 1-alpha)*, *TUBULIN (Beta tubulin)*, *GADPH (Glyceraldehyde-3-phosphate dehydrogenase)*, *APN (Aminopeptidase N)*, *ALP (Alkaline phosphatase)*, and *SOD (Superoxide dismutase)*. The retrieved sequences were blasted using a BLASTx search of each sequence in the NCBI database to reconfirm their annotations. Primer3 software was used to design the primers for RT-qPCR, which aimed to amplify 100–150 bp amplicons of the individual gene. (Appendix A). SYBR Premix Ex Taq II (Tli RNaseH Plus) (Clontech) was used on a LightCycler 96 (Roche Molecular Systems, Inc., Pleasanton, CA, USA). The RT-qPCR reaction performed in triplicates contained 0.2 µL of 10 mM of gene-specific primers and 1 µL of 1:10 diluted cDNA per reaction. To ensure that the cDNA reagents did not impair the efficiency, absolute quantification was also carried out with three 10-fold serial dilutions. The amplification cycle involved initial denaturation at 95 °C for the 30 s, followed by 40 cycles of 95 °C for 5 s, and 55 °C for 10 s. The standard curve and melting curve analysis confirmed the reliability of the gene amplification by RT-qPCR.

### 4.5. Expression Stability Analysis of Reference Genes

The most efficient reference gene was selected from the candidate genes based on their expression stability. Using the algorithms geNorm, Normfinder, and Bestkeeper, as well as RefFinder analysis, the expression stability of genes was evaluated to identify the most potential reference genes for the accurate normalization of target genes expression in all the developmental stages. The most consistently expressed gene was determined by performing a pair-wise variation (V) value after calculating the gene expression stability value (M). NormFinder calculates the expression stability based on intra-group variance and wherever required also considers intergroup variances. BestKeeper uses Ct values for stability analysis and these were acquired from the LightCycler 96 software. Further, the most stable gene is determined by analyzing the standard deviation, p values, index, and correlation coefficient of each gene. RefFinder calculates the geometric mean of the weights assigned to each gene while considering the rank provided by each statistical algorithm to arrive at the final ranking.

### 4.6. Identification of Functional Candidate Genes and Phylogenetic Analysis

To identify the protein sequences associated with cytochrome P450 (CYP450), cathepsin, and cadherin in *E. vittella*, the transcriptome was searched for the genes by constructing a profile Hidden Markov Model (pHMM) for the proteins in the question using HMMER 3.2.1 (http://www.hmmer.org/, accessed on 2 September 2022). The transcript sequences were translated into six possible reading frames using TransDecoder v.2.0.1, followed by searching the translated transcriptome data with each of the profile HMMs obtained from Pfam (accession no.: PF00067, PF00112, and PF08266 for CYP450, cathepsin, and cadherin, respectively) using an HMMER search with default parameters. The identified genes were reconfirmed through a BLASTx search against the NCBI nr database. Consequently, the redundant sequences were removed using ClustalW. Multiple alignments of the protein sequences were generated along with the representative genes of the proximal species using ClustalW. To infer the phylogenetic relationships, the phylogenetic trees were created using the Maximum Likelihood Tree method in the MEGA package (version 10.1.7). The deduced amino acid sequences were subjected to phylogenetic analysis to reduce the effect of nucleotide compositional bias. The constructed tree was further visualized in FigTree (http://tree.bio.ed.ac.uk/software/figtree/, accessed on 2 September 2022) to allocate the sequences to their respective clades. The analysis was performed individually for all the three gene families of *E. vittella*.

### 4.7. Validating Transcriptomic Data across Development Stages Using RT-qPCR

The functionally annotated transcriptome data of *E. vittella* allowed for the identification of various genes involved in controlling the insect′s essential physiological functions, including detoxification, digestion, defense, and signaling. (Appendix A). A total of seven developmental stages were taken, i.e., first, second, third, fourth, and fifth instars, male and female pupae, and male and female adults (Figure 1). Adult males and females were recognized at the pupal stage by observing a knob-like structure at the anterodorsal end of the cocoon [88]. Approximately 15–20 first instar larvae, fifteen second instar larvae, nine third instar larvae, three fourth instar larvae, three fifth instar larvae, three adult females, and three adult males were taken for RNA isolation and further experimental procedures to validate the expression of target genes across developmental stages. Three biological replicates of each sample were taken and stored at −80 °C for further use. The total RNA was extracted from each development stage of the insect using the RNAiso Plus (Takara) and quantified using a NanoDrop spectrophotometer. cDNAs were synthesized using the RevertAid First Strand cDNA Synthesis Kit (Thermo Fisher Scientific, Waltham, MA, USA) from 1 µg of RNA, following the manufacturer′s instructions. The cDNA samples used in the reaction setup were diluted tenfold, and the same cDNA was used to quantify the differentially expressed genes across the developmental stages. The expression was validated by RT-qPCR reaction using SYBR^®^ Premix Ex TaqTMII, Takara, according to the manufacturer′s guidelines, and the reaction was conducted in the Light cycler System (Roche life sciences, Mannheim, Germany).

### 4.8. RNAi Experiments

#### 4.8.1. Preparation of dsRNA

Target genes for RNAi were amplified using cDNA and used as a template using gene-specific primers (Appendix A). To amplify templates for target dsRNA, PCR was performed in a total volume of 25 μL using 1 μL of cDNA, 12.5 μL of 2 × Master Mix (Emerald, Takara), and 0.5 μL of 10 μM for gene-specific primers. The reaction was amplified with the following PCR conditions: initial denaturation was carried out for 5 min at 95 °C, then 35 cycles of 45 s at 95 °C, 56 °C, and 72 °C were performed. The process was then extended for 10 min at 72 °C. The amplified products were resolved on 1% agarose gel followed by PCR product or gel cleanup using NucleoSpin^®^ Gel and a PCR Clean-up kit (Takara Bio, USA). The quantification of purified PCR products was performed using a Spectrophotometer (Eppendorf) and products were stored at −20 °C until further processing. The double-stranded RNA (dsRNA) of target genes and green fluorescence protein (GFP, control) was synthesized by using a MEGAscript™ RNAi Kit. The integrity of the dsRNA was analyzed on 1% agarose gels, and the concentration was determined by a Spectrophotometer (Eppendorf, Pocklington, York, UK).

#### 4.8.2. dsRNA Stability Studies

Hemolymph degradation studies were performed to find out the effect of nucleases on orally fed naked dsRNA. So, hemolymph was extracted from fourth instar larvae and kept in a 1.5 mL Eppendorf containing Phenylthiourea dissolved in 1X PBS and kept on ice to prevent melanisation. It was further centrifuged at 12,000 rpm for 10 min at 4 °C. The supernatant was collected and used to perform degradation studies. Estimation of the protein concentration was performed with the help of Bradford′s assay. Based on the protein concentration of hemolymph, serial dilutions were prepared. Different dilutions were incubated with 500 ng dsGFP for 1, 3, 5, and 10 h. The degradation of dsRNA was estimated by running the dilutions on the agarose gel [15].

#### 4.8.3. Preparation of dsRNA–CQD (Carbon Quantum Dots) Nanoconjugates

CQD nanoparticles were prepared as per methodology [22]; for preparation, 9 mL of PEG-200 (polyethylene glycol—M. W. 200) was combined with 3 mL of water followed by the addition of 100 mg of PEI (polyethylenimine) in 2 mL of deionized water. A faint golden yellow coloration emerged after 3 min of 800 W microwave heating of the combination. The mixture was allowed to cool and was followed by mixing 100 μL of PEG-PEI solution and 100 μL of sodium sulfate solution with 40 μg of target gene dsRNA. The mixture was incubated overnight at 4 °C followed by centrifugation at 12,000 rpm for 10 min. The obtained pellet was dissolved in 20 μL double distilled water.

#### 4.8.4. Preparation of Chitosan Coated dsRNA Nanoconjugates

Chitosan dsRNA nanoparticles were prepared from 1μg of chitosan (Merck (Sigma–Aldrich), Darmstadt, Germany) added to 100 μL 0.1 M sodium acetate buffer followed by mixing it with 100 μL of 50 mM sodium sulfate buffer having 1 μg of dsRNA dissolved in it as per the earlier described methodology [22]. To allow for the formation of nanoparticles, the mixture was heated at 55 °C for 1 min, immediately vortexed for 30 to 60 s at high speed followed by an hour of incubation at room temperature. For 10 min, the mixture was centrifuged at 13,000 rpm. The pellet was air-dried after removing the supernatant and dissolved in RNase and DNase-free ddH_2_O. The gel retardation assay was performed to verify the dsRNA–chitosan conjugation and to evaluate the ideal concentration of chitosan and lipofectamine necessary to complete the conjugation process.

#### 4.8.5. Preparation of Lipofectamine dsRNA Conjugate

The preparation of the dsRNA Lipofectamine conjugate was conducted as per protocol [94]. A total of 4 µL of the transfection reagent (Thermo Scientific Turbofect cat# R0532) was diluted by adding 4 µL of 5% glucose solution. The dsRNA was also diluted using a 5% glucose solution. The diluted liposome reagent solution was immediately added to the diluted dsRNA solution all at once. The solution was vortexed briefly to mix them and finally incubated for 15 min at 25 °C.

### 4.9. Characterization of Nanoparticles

The transmission electron microscopy (TEM) study was performed to compare the size and shape of nanoparticles before and after conjugation with dsRNA. The nanoparticle suspension was prepared by bath sonication for 15 min. The sonicated suspension (10 µL) was placed on the carbon-coated copper grid (200 mesh, Tedpella, CA, USA). These grids were dried at room temperature for approximately 4 h and then viewed under TEM (Hitachi H-7650, Japan) operated at an accelerating voltage of 80 Kv.

### 4.10. Insect Bioassays and dsRNA-Mediated Gene Knockdown

The insects were subjected to 12h of starvation before feeding on an artificial diet. The naked dsRNA (10 μg) and Chitosan, CQD, and lipofectamine dsRNA nanoconjugates against different target genes (*CAD*, *AMN*, *AMY*, *CS*, V-*ATPase*, *JHAMT*) and control gene *Green fluorescent protein* (dsGFP) were incorporated in the artificial diet. A total of six biological replicates representing an individual 4th instar were fed with dsRNA orally for 72 h. Post 72 h of feeding access to the dsRNA diet mixture, the total RNA from each individual was isolated using RNAiso Plus (Takara) from insect samples followed by reverse transcription into cDNA as described earlier in these studies. The template cDNA was diluted ten-fold and 1μL of it was added to RT-qPCR with 0.1 μL of each primer (10 μM), and 5 μL of SYBR^®^ Premix Ex TaqTM II in a total 10 μL PCR reaction. The RT-qPCR conditions consisted of one cycle at 95 °C for 30 s, 40 cycles at 95 °C for 5 s, and 30 cycles at 60 °C. The LightCycler^®^ 96 Real-Time PCR System (Roche Applied Science, Charles Avenue, West Sussex, UK) was used to conduct the reactions. Using the Ct technique [95], the relative quantification of genes was carried out, and the significance level was assessed using the Student′s *t*-test (*p* = 0.05). *GAPDH* was used as a housekeeping gene for normalization and finally, the relative expression was compared to dsGFP-fed controls.

### 4.11. Expression of Core RNAi Genes with Gene Knockdown Samples

The relative expression of core components of the RNAi (*RNA binding protein*, *dsRNAse*, *Dicer*, and *Piwi*) was calculated in *E. vittella*. A total of three groups of insects, with six biological replicates in each group, were taken. The first group was fed with chitosan-coated dsGFP (10 µg) in a synthetic diet. Similarly, the second group was fed with non-coated dsGFP (10 µg), and the third group, considered as control, was fed with the diet only.

## 5. Conclusions

The first assembled transcriptome of *E. vittella* can be used as a molecular resource for future studies in this insect. This study identified candidate reference genes and validated them across different development stages using various algorithms, and these data are beneficial to researchers for gene expression studies in this insect. In addition, this study identified the functional candidate genes related to *juvenile hormone* (*JH*) and *ecdysteroid* biosynthetic pathways, and homologs of cytochrome P450 enzymes, cathepsins, and cadherins that can be used as potential RNAi targets to manage this insect pest. By improving the RNAi efficacy, coupling dsRNA with nanoparticles such as chitosan, lipofectamine, and CQD would help and make it easier to see the precise effects of specific gene knockdowns on *E. vittella*. On the other hand, future pest management techniques based on dsRNA-nano conjugate sprays may enhance the RNAi through the knockdown of target genes. Although, strategies for insecticide distribution could be modified to serve as RNAi trigger delivery systems, despite certain restrictions on cost and environmental concerns. This would hasten the transition of RNAi technology from the lab to the field. We thus conclude that the nanoshield dsRNA delivery technology may aid in functional genomics studies as well as in developing strategies for the management of *E. vittella* populations.

## Figures and Tables

**Figure 1 ijms-24-09161-f001:**
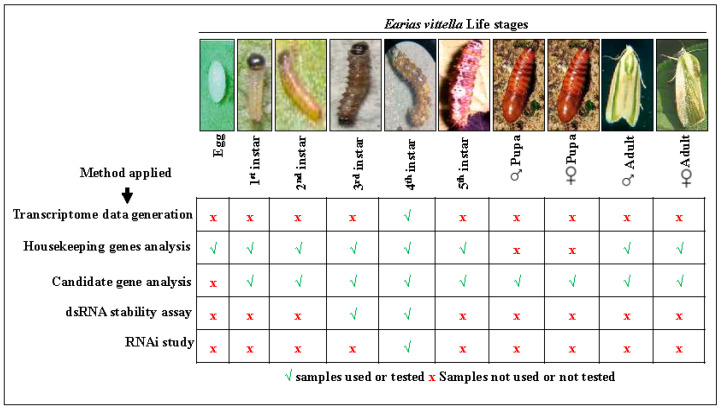
Experimental design and samples used for transcriptome data generation, housekeeping genes analysis, candidate gene analysis, dsRNA stability assay, and RNAi study. The *E. vittella* 4th instar larvae were used for transcriptome data generation. For the dsRNA stability assay, hemolymph was collected from 3rd or 4th instar larvae. For the RNAi study, 4th instar larvae were used for naked dsRNA and nanoparticle-shielded dsRNA feeding.

**Figure 2 ijms-24-09161-f002:**
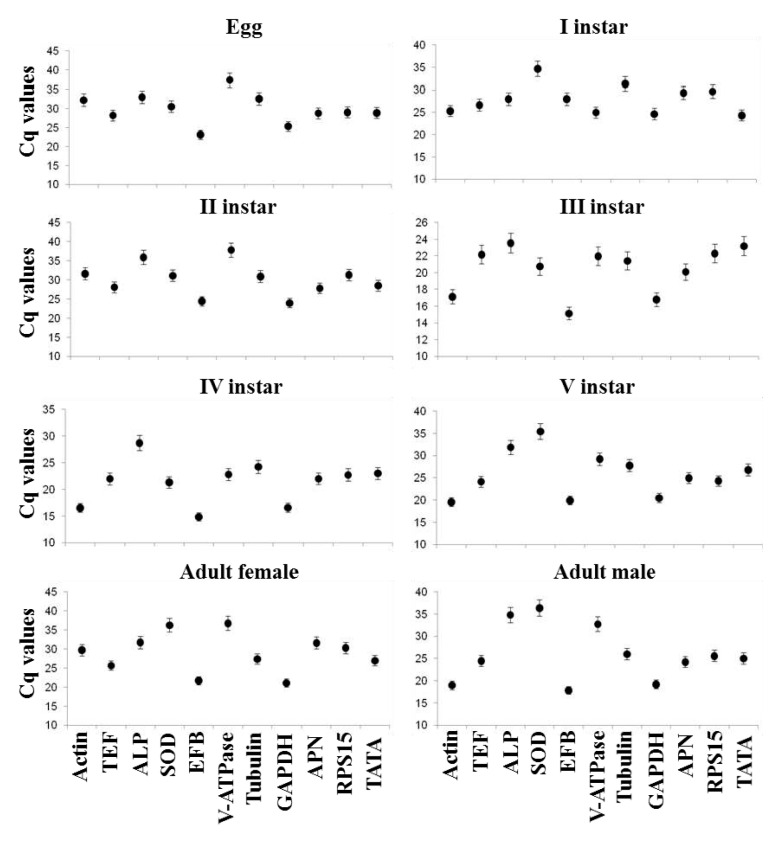
Cq-values-based expression profile of eleven candidate reference genes from *E. vittella* across various developmental stages. The error bars represent SE ± mean.

**Figure 3 ijms-24-09161-f003:**
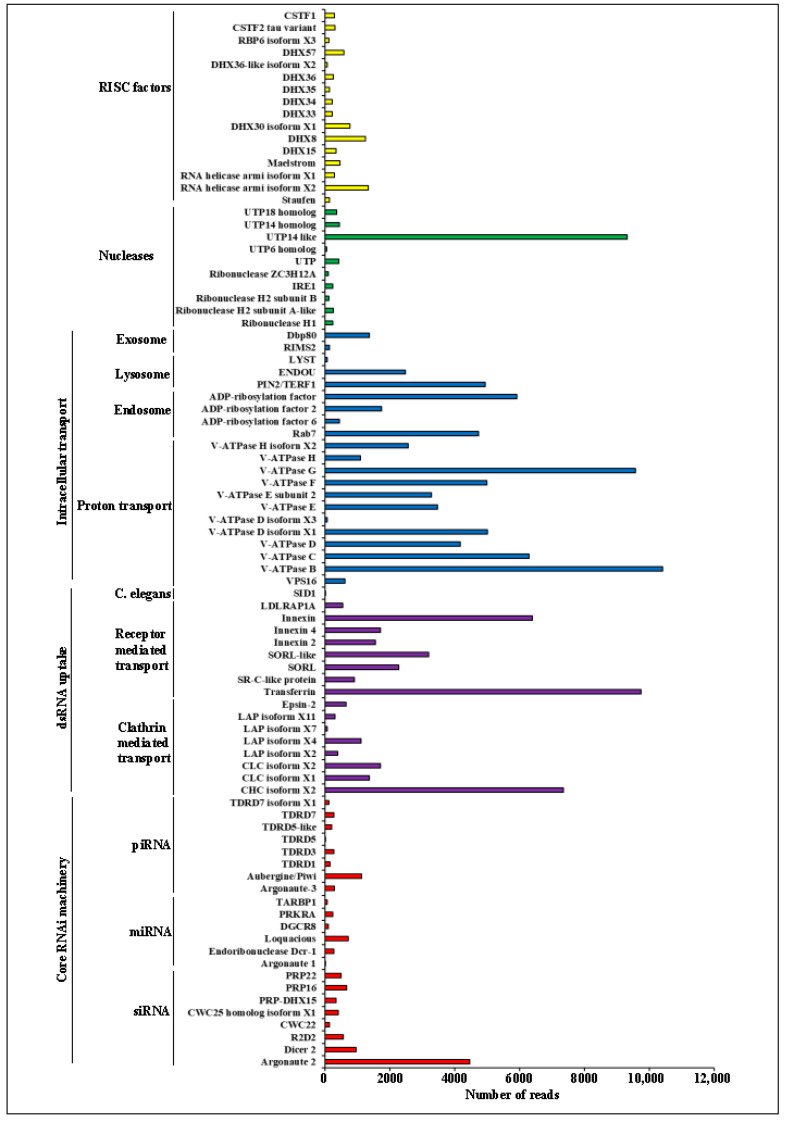
RNAi machinery genes identified in *E. vittella*. X-axis represents the number of reads and Y-axis represents the gene names. The various subcategories of RNAi machinery genes are represented by the color bars.

**Figure 4 ijms-24-09161-f004:**
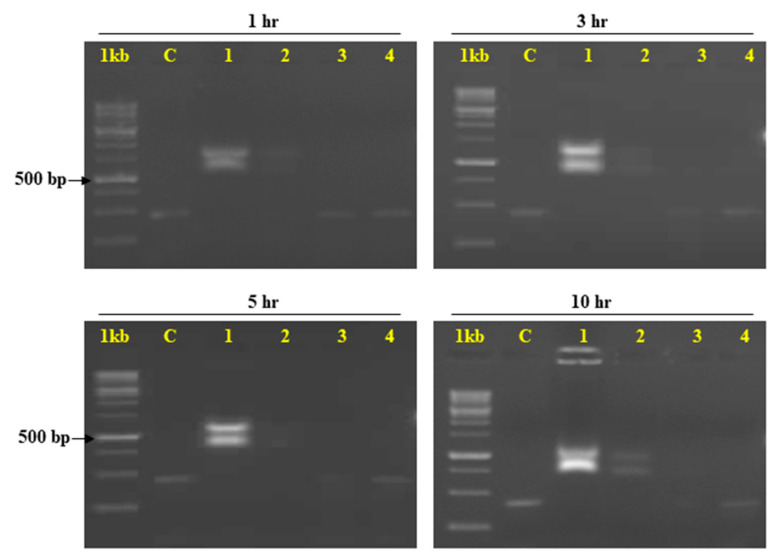
Agarose gel electrophoresis analysis of dsRNA of *E. vittella* after exposing dsGFP to various concentrations of hemolymph (followed by incubation at room temperature for 1, 3, 5, and 10 h). 1 kb—1 kb ladder, C—control (only dsGFP), 1—0.97 µg/mL, 2—0.12 µg/mL, 0.015 µg/mL, 0.002 µg/mL.

**Figure 5 ijms-24-09161-f005:**
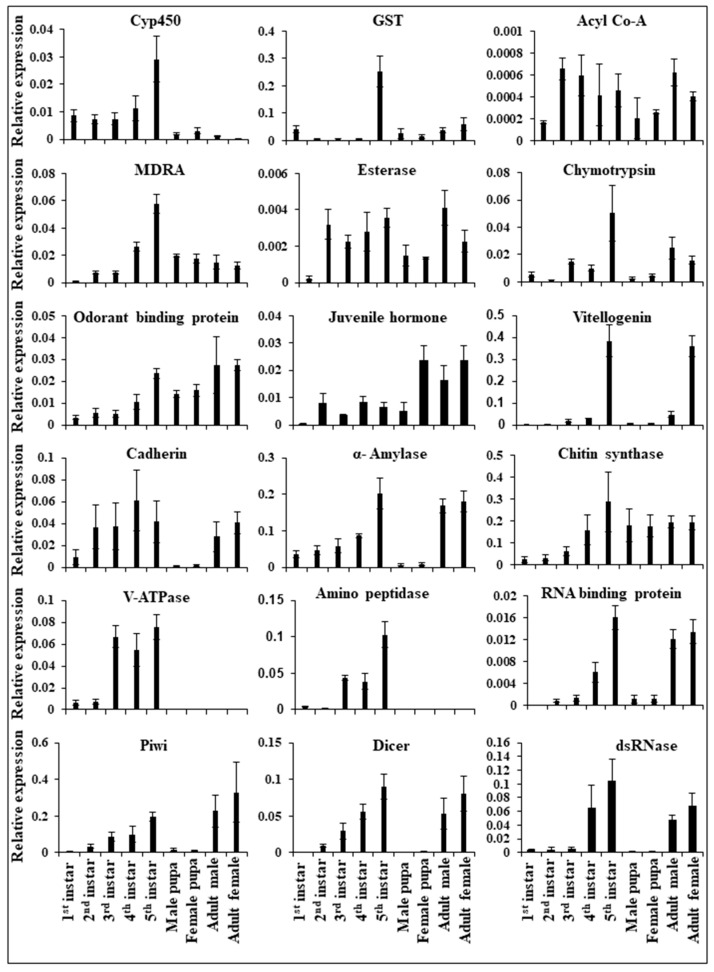
Expression of functional genes in different developmental stages of *E. vittella* based on total reverse transcription of total RNA into cDNA followed by quantification of mRNA levels using gene-specific primers in RT-qPCR. The expression of target genes has been normalized using GAPDH as a reference gene. The average relative mRNA level and standard error (*N* = 3). A single variable′s mean was compared using a one-tailed *t*-test. X-axis represents developmental stages, Y-axis represents the relative expression.

**Figure 6 ijms-24-09161-f006:**
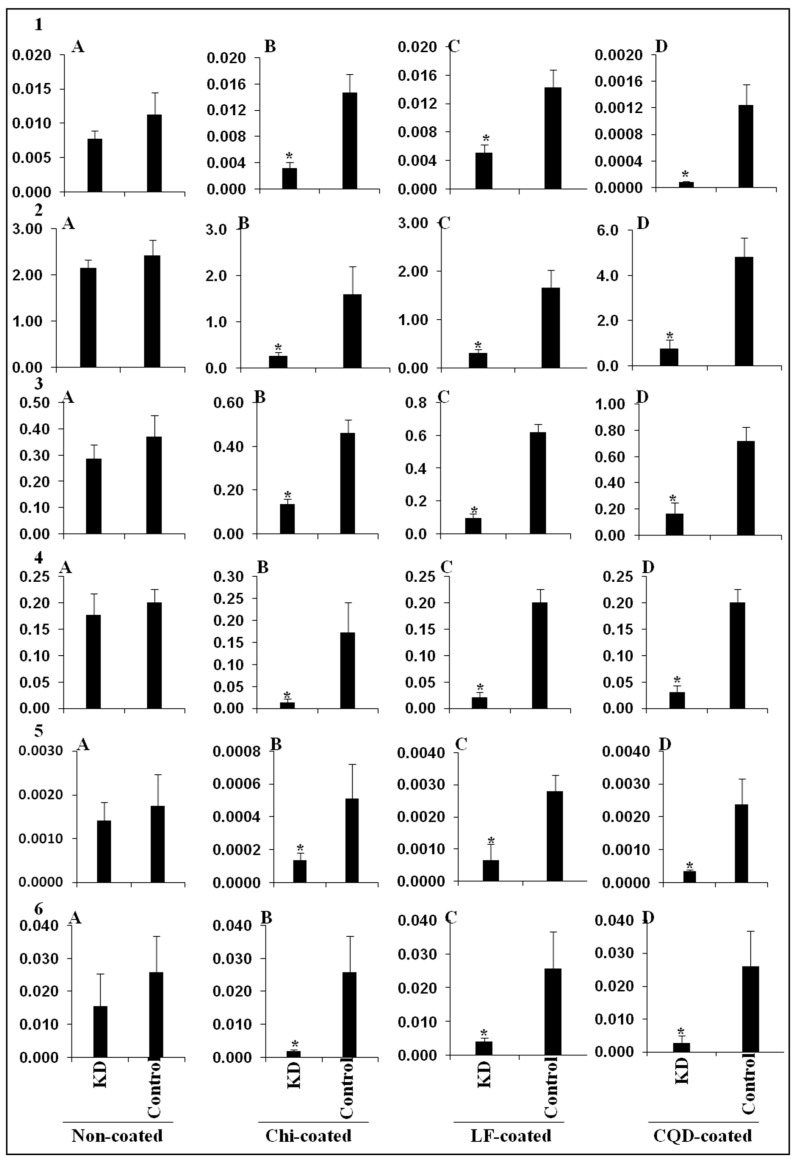
Comparative RT-qPCR analysis to evaluate the knockdown (KD) of the target gene using nanoparticle-conjugated dsRNA and non-coated dsRNA fed to *E. vittella*. Ten micrograms of the target genes dsRNA and dsGFP (control) were fed with *E. vittella* fourth instar larvae. Extraction of total RNA was conducted on the fifth-day post-feeding of dsRNA. The gene expression of targets was quantified using gene-specific primers after reverse transcription of total RNA into cDNA with delta-CT. The expression of target genes was normalized using GAPDH as a reference gene. The average relative mRNA level and standard error (*N* = 3). A single variable′s mean was compared using a one-tailed *t*-test. X-axis represents the treatments [A—Non-coated; B—Chitosan-coated; C—Lipofactamine-coated; D—CQD-coated], Y-axis represents the relative expression [1—*Cadherin*; 2—*Aminopeptidase*; 3—*Alpha-amylase*, 4—*Chitin synthase*; 5—*V-ATPase*; 6—*Juvenile hormone methyl transferase*]”. KD is an acronym for gene knockdown. * indicates significant difference in target gene expression compared to control based on Student’s *t*-test (*p* = 0.05).

**Figure 7 ijms-24-09161-f007:**
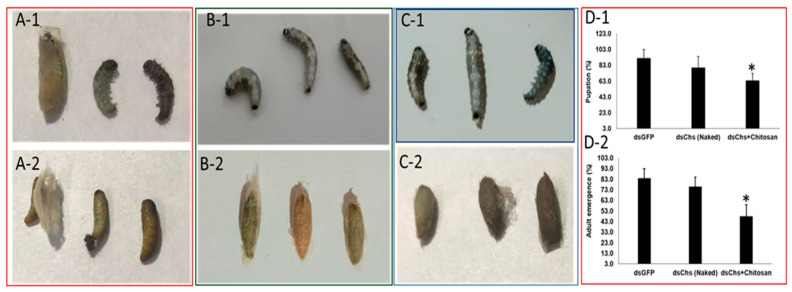
Comparative evaluation of the impact of knockdown of Chitin synthase with chitosan-coated and non-coated dsRNA on pupation and adult emergence of *E. vittella.* The 4th instar larvae were fed with 20 µg dsChitin synthase (dsChs) two times with a gap of 72 h after the first feeding. The larvae were observed till adult emergence. (**A-1**): Larval deformities with dsChs + Chitosan; (**A-2**): pupal phenotype without hairy tuft in dsChs + Chitosan; (**B-1**): larvae fed with naked dsChs (normal development); (**B-2**): pupae in naked dsChs (normal development with hairy tuft); (**C-1**): larvae fed with dsGFP (normal development); (**C-2**): pupae in naked dsGFP (normal development with hairy tuft); (**D-1**): percent pupation; (**D-2**): adult emergence. Error bars represent SE ± M based on five replicates each with five 4th instar larvae. * Indicates significance using Student′s *t*-test differences compared to GFP control (*p* = 0.05).

**Figure 8 ijms-24-09161-f008:**
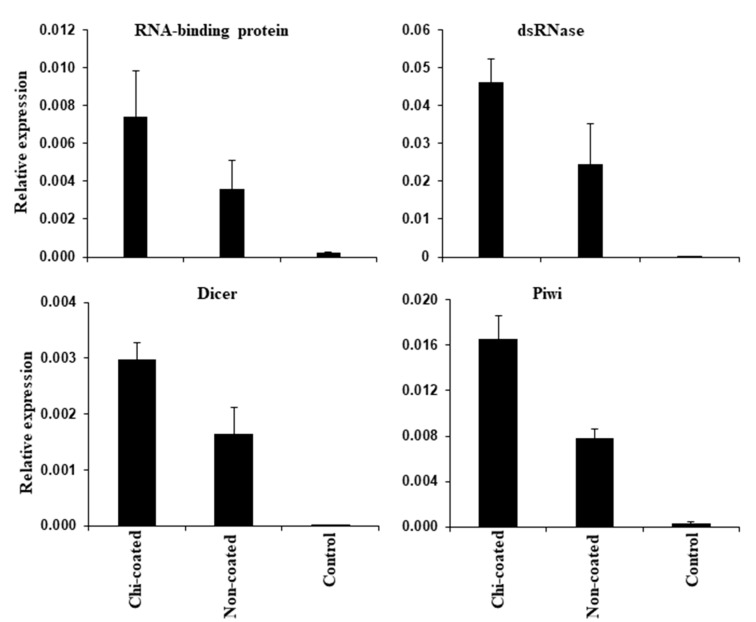
Expression of core RNAi machinery genes in gene knockdown samples of *E. vittella*. Total RNA was isolated from dsRNA-treated samples and reverse transcribed into cDNA, and the cDNA was amplified using gene-specific primers to quantify the mRNA levels of the target genes in RT-qPCR. The GAPDH gene was used to normalize expressions. The mean of relative mRNA levels and SE (*N* = 3). A single variable′s mean was compared using a one-tailed *t*-test. X-axis represents the dsRNA-treated samples, Y-axis represents the relative expression.

**Table 1 ijms-24-09161-t001:** Evaluation of potential reference genes by using geNorm, NormFinder, BestKeeper, ∆CT, and RefFinder analyses according to their stability value *.

Gene Name	Egg	1st Instar	2nd Instar	3rd Instar	4th Instar	5th Instar	Adult Female	Adult Male	All Stages
geNorm									
*ACTIN*	10	10	9	2	2	7	4	9	8
*TEF*	2	2	7	7	1	8	3	1	2
*ALP*	5	8	8	9	9	3	8	1	7
*SOD*	9	3	10	4	10	1	6	4	10
*EFB*	1	9	2	1	5	5	5	7	5
*V-ATPase*	6	4	6	3	6	10	1	10	9
*TUBULIN*	7	1	4	5	7	1	7	8	1
*GADPH*	3	7	5	6	3	6	2	2	1
*APN*	8	1	1	10	8	9	9	5	4
*RPS15*	1	6	1	8	4	4	1	6	3
*TATA*	4	5	3	1	1	2	10	3	6
**NormFinder**									
*ACTIN*	11	11	10	1	2	3	4	10	9
*TEF*	1	2	8	7	6	4	5	1	3
*ALP*	9	8	9	10	8	9	9	2	8
*SOD*	10	1	11	3	11	5	10	8	11
*EFB*	4	10	3	8	4	2	7	6	7
*V-ATPase*	6	4	2	2	1	11	1	11	10
*TUBULIN*	7	6	1	4	7	8	6	9	5
*GADPH*	5	7	7	5	5	1	2	4	2
*APN*	8	5	6	11	10	10	8	3	5
*RPS15*	3	9	4	9	9	7	3	5	1
*TATA*	2	3	5	6	3	6	11	7	6
**BestKeeper**									
*ACTIN*	9	2	10	4	2	2	5	10	11
*TEF*	5	5	2	8	3	3	2	4	2
*ALP*	10	8	6	11	10	8	11	6	6
*SOD*	11	6	11	5	11	6	10	9	9
*EFB*	7	11	7	3	6	4	9	2	8
*V-ATPase*	1	7	4	1	4	11	4	3	10
*TUBULIN*	3	10	1	6	8	10	3	11	5
*GADPH*	6	1	3	7	5	1	1	5	4
*APN*	2	9	8	10	9	9	8	7	7
*RPS15*	8	3	5	9	7	7	6	1	3
*TATA*	4	4	9	2	1	5	7	8	1
**∆CT**									
*ACTIN*	11	11	10	2	1	6	4	10	9
*TEF*	1	1	8	8	5	8	5	1	3
*ALP*	9	9	9	10	9	9	10	2	8
*SOD*	10	2	11	3	11	3	9	6	11
*EFB*	3	10	2	7	6	2	7	8	7
*V-ATPase*	6	6	6	4	3	11	1	11	10
*TUBULIN*	7	5	1	1	8	5	6	9	4
*GADPH*	5	7	7	5	4	1	3	3	1
*APN*	8	4	4	11	10	10	8	4	5
*RPS15*	4	8	3	9	7	7	2	7	2
*TATA*	2	3	5	6	2	4	11	5	6
**Comprehensive**									
*ACTIN*	11	9	10	1	2	4	5	11	9
*TEF*	1	1	8	8	4	7	4	1	3
*ALP*	9	10	9	10	9	9	10	2	8
*SOD*	10	2	11	6	11	2	9	8	11
*EFB*	3	11	3	5	6	3	7	6	7
*V-ATPase*	5	7	5	2	3	11	1	9	10
*TUBULIN*	8	5	1	4	8	6	6	10	4
*GADPH*	6	6	6	7	5	1	2	3	1
*APN*	7	3	4	11	10	10	8	5	6
*RPS15*	4	8	2	9	7	8	3	4	2
*TATA*	2	4	7	3	1	5	11	7	5

* The numbers represent gene rank in terms of the suitable reference gene in each respective growth stage based on the stability value obtained using various algorithms and overall comprehensive analysis.

## Data Availability

All the data relevant to the study has been provided in the Manuscript and Appendix A.

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
