# Peer review of "Nanoparticle-Shielded dsRNA Delivery for Enhancing RNAi Efficiency in Cotton Spotted Bollworm Earias vittella (Lepidoptera: Nolidae)"

_ijms, 2023, doi:10.3390/ijms24119161_

Round 1
Reviewer 1 Report
This manuscript describes a RNAi treatment-based gene silence for cotton spotted bollworm through nanoparticle-mediated delivery. This author identified candidate reference genes and verified their applicability to the pest gene expression inhibition that leads to pest mortality. In my opinion, this work is of good value and technological importance, and may have implications in treatment of cotton diseases by bionanotechnology. But no sufficient evidence was provided to validate the application of the treatment. To improve the quality and scientific impact of this work, I recommend the rejection of the paper due to the occurrence of the following major issues, while the author may resubmit it if they could solve these issues.
1. No any characterization results were shown for various nanoparticles before and after PEI modification and RNA loading. This is weird! Without the data, no trust for the results.
2. No results were given in the manuscript regarding the efficient loading of RNA onto nanoparticles. There is no trust for the results without the data.
3. The author also needs to evidence the entrance of the the nanoparticle-RNA complex into the worm organs and even cells, so that people will believe that they can exert the function of gene expression interference.
4. What is the mechanism for the shielding effect of nanoparticles on RNA? Even it is difficult to provide evidence, the author should make a proposal for it.
5. Is the pest mortality attributable to RNAi-inhibited gene expression? The author should provide control experiment results to exempt the effect of nanoparticles.
6. I recommend the author to add a schematic illustration to highlight the logic, concept and mechanism of this study, so that people can capture the main point quickly, and the work will make a greater impact in the field.
Author Response
Open Review 1 Comments and Suggestions for Authors
This manuscript describes a RNAi treatment-based gene silence for cotton spotted bollworm through nanoparticle-mediated delivery. This author identified candidate reference genes and verified their applicability to the pest gene expression inhibition that leads to pest mortality. In my opinion, this work is of good value and technological importance, and may have implications in treatment of cotton diseases by bionanotechnology. But no sufficient evidence was provided to validate the application of the treatment. To improve the quality and scientific impact of this work, I recommend the rejection of the paper due to the occurrence of the following major issues, while the author may resubmit it if they could solve these issues.
Author’s response: We are appreciate reviewers for encouraging comments on the present work. Yes, we agree with the reviewer comments and in response to that we have performed some of the studies which will answer the concerns raised by the reviewer. Besides that we want to apprise that our group has tested and used nanoparticles (Chitosan, CQD, and Lipofectamine) in RNAi insensitive and moderately RNAi responsive insect species like Spodoptera frugiperda, Bemisia tabaci, and Aedes aegypti and published the work in journals of good repute. So the protocols used in these studies have been well established and in context to preparation of nanoparticles , their characterization as well as characterization of dsRNA-nanocomplexes. So, in regard we have cited our previous research articles in respective places and used directly these nanoparticles as well as dsRNA-nanocomplexes for our study, however we have supplemented information on characterization of these dsRNA-nanocomplexes based on reviewer suggestions.
Reviewer comment 1: No any characterization results were shown for various nanoparticles before and after PEI modification and RNA loading. This is weird! Without the data, no trust for the results.
Author’s response: It is well taken that characterization of the nanoparticles and dsRNA-nanocomplexes needs to be done, however our group has already performed these pre-requisites and published good articles about the used nanoparticles (Chitosan, CQD, and Lipofectamine) used in this study and also we have a well-established protocol for used nanoparticle preparation as well as dsRNA-nanocomplexes formation. In our previous reports, we showed clear data (DLS, TEM, and Gel retardation assay) about nanoparticles and dsRNA-nanocomplexes preparation, and characterization. However keeping in view the suggestion and to make our study practically sound we had characterized the chitosan and CQD- dsRNA nanoconjugates using transmission electron microscopy to study about size, shape and formation of complex. The relevant methodology for the same techniques employed for characterization has been added in the manuscript and results has been given as Supplementary Fig. S11
In previous studies using these dsRNA-nanocomplexes we showed very good mortality and gene knockdown in lepidopteran and hemipteran insects. Some of relevant previous articles cited in this MS are given below.
References:
- Gurusamy, D.; Mogilicherla, K.; Palli, S.R. Chitosan nanoparticles help double-stranded RNA escape from endosomes and improve RNA interference in the fall armyworm, Spodoptera frugiperda. Arch. Insect Biochem. Physiol.2020, 104, e21677, doi:10.1002/arch.21677.
- Kaur, R.; Gupta, M.; Singh, S.; Joshi, N.; Sharma, A. Enhancing RNAi Efficiency to Decipher the Functional Response of Potential Genes in Bemisia tabaciAsiaII-1 (Gennadius) Through dsRNA Feeding Assays. Front. Physiol.2020, 11, 123, doi:10.3389/fphys.2020.00123.
- Gurusamy, D., Mogilicherla, K., Shukla, J.N. and Palli, S.R., 2020. Lipids help double‐stranded RNA in endosomal escape and improve RNA interference in the fall armyworm, Spodoptera frugiperda. Archives of Insect Biochemistry and Physiology, 104(4), p.e21678.
- Das, S.; Debnath, N.; Cui, Y.; Unrine, J.; Palli, S.R. Chitosan, Carbon Quantum Dot, and Silica Nanoparticle Mediated dsR-NA Delivery for Gene Silencing in Aedes aegypti: A Comparative Analysis. ACS Appl. Mater. Interfaces2015, 7, 19530–19535, doi:10.1021/acsami.5b05232
Reviewer comment 2: No results were given in the manuscript regarding the efficient loading of RNA onto nanoparticles. There is no trust for the results without the data.
Author’s response: Thanks for raising this critical point so as per suggestion the agarose gel image showing gel retardation assay of chitosan, lipofectamine and CQD-dsRNA nanoconjugate has been added in and Supplementary Fig. S12. It has been well established in the previous studies and pertinent to mention that dsRNA-nanoparticles loaded on the agarose gel stuck in the wells and cannot be visualized thereof. To prove the dsRNA-nanoparticles binding we have conjugated the various nanoparticles at serial concentration with dsRNA to prove that a threshold concentration of nanoparticles is required to hold the dsRNA and when the concentration falls below this level the dsRNA is released and can be visualized in the agarose gel. The gel retardation assay proves that nanocarrier hold/ bind to the dsRNA (Supplementary Figure S12). To further support this, it is mentioned that our group has published the used nanoparticles (Chitosan, CQD, and Lipofectamine and their preparation and demonstrated the loading of dsRNA into and this has been further proved through good mortality and gene knockdown in lepidopteran and hemipteran insects. Some of these articles have been cited in the MS
References:
- Gurusamy, D.; Mogilicherla, K.; Palli, S.R. Chitosan nanoparticles help double-stranded RNA escape from endosomes and improve RNA interference in the fall armyworm, Spodoptera frugiperda. Arch. Insect Biochem. Physiol.2020, 104, e21677, doi:10.1002/arch.21677.
- Kaur, R.; Gupta, M.; Singh, S.; Joshi, N.; Sharma, A. Enhancing RNAi Efficiency to Decipher the Functional Response of Potential Genes inBemisiatabaciAsiaII-1 (Gennadius) Through dsRNA Feeding Assays. Front. Physiol.2020, 11, 123, doi:10.3389/fphys.2020.00123.
- Gurusamy, D., Mogilicherla, K., Shukla, J.N. and Palli, S.R., 2020. Lipids help double‐stranded RNA in endosomal escape and improve RNA interference in the fall armyworm, Spodoptera frugiperda. Archives of Insect Biochemistry and Physiology, 104(4), p.e21678.
- Das, S.; Debnath, N.; Cui, Y.; Unrine, J.; Palli, S.R. Chitosan, Carbon Quantum Dot, and Silica Nanoparticle Mediated dsR-NA Delivery for Gene Silencing in Aedes aegypti: A Comparative Analysis. ACS Appl. Mater. Interfaces2015, 7, 19530–19535, doi:10.1021/acsami.5b05232.
Reviewer comment 3: The author also needs to evidence the entrance of the nanoparticle-RNA complex into the worm organs and even cells, so that people will believe that they can exert the function of gene expression interference.
Author’s response: The feeding of the dsRNA-nanoparticles mixtures has been done through semi-synthetic diet. Our group’s previous studies have demonstrated and established the uptake of dsRNA in the body and cells of lepidopteran as well as many other insect orders through P32 labelled dsRNA and its recovery thereof from the total RNA of insect/ insect tissues as well as the translocation of dsRNA in the insect cells through Confocal microscopy (Shukla et al 2016, Singh et al 2017). To support this in case of our current studied with Earias vittella, we have fed the food grade red dye to mixed with dSRNA -nanoconjugates to the insect followed by dissection of midgut and observation of fecal matter compared to control insects to prove the entry of nanoparticles and dsRNA in the insect body (Supplementary Figure S 13). Also, we referred to our previous articles in the respective places and some of the relevant studies in support to our statements are mentioned below:
References:
- Shukla, J.N.; Kalsi, M.; Sethi, A.; Narva, K.E.; Fishilevich, E.; Singh, S.; Mogilicherla, K.; Palli, S.R. Reduced stability and intracellular transport of dsRNA contribute to poor RNAi response in lepidopteran insects. RNA Biol. 2016, 13, 656–669, doi:10.1080/15476286.2016.1191728.
- Singh, I.K.; Singh, S.; Mogilicherla, K.; Shukla, J.N.; Palli, S.R. Comparative analysis of double-stranded RNA degradation and processing in insects. Rep. 2017, 7, 17059, doi:10.1038/s41598-017-17134-2
- Gurusamy, D.; Mogilicherla, K.; Palli, S.R. Chitosan nanoparticles help double-stranded RNA escape from endosomes and improve RNA interference in the fall armyworm, Spodoptera frugiperda. Arch. Insect Biochem. Physiol.2020, 104, e21677, doi:10.1002/arch.21677.
- Kaur, R.; Gupta, M.; Singh, S.; Joshi, N.; Sharma, A. Enhancing RNAi Efficiency to Decipher the Functional Response of Potential Genes inBemisiatabaciAsiaII-1 (Gennadius) Through dsRNA Feeding Assays. Front. Physiol.2020, 11, 123, doi:10.3389/fphys.2020.00123.
- Gurusamy, D., Mogilicherla, K., Shukla, J.N. and Palli, S.R., 2020. Lipids help double‐stranded RNA in endosomal escape and improve RNA interference in the fall armyworm, Spodoptera frugiperda. Archives of Insect Biochemistry and Physiology, 104(4), p.e21678.
- Das, S.; Debnath, N.; Cui, Y.; Unrine, J.; Palli, S.R. Chitosan, Carbon Quantum Dot, and Silica Nanoparticle Mediated dsR-NA Delivery for Gene Silencing in Aedes aegypti: A Comparative Analysis. ACS Appl. Mater. Interfaces2015, 7, 19530–19535, doi:10.1021/acsami.5b05232.
Reviewer comment 4: What is the mechanism for the shielding effect of nanoparticles on RNA? Even it is difficult to provide evidence, the author should make a proposal for it.
Author's response: The primary factor influencing the stability of dsRNA in insects is nucleases enzymes. The enzymatic activity is higher in gut of lepidopterans. Therefore in this study (Fig. 4) and previous studies it has been proved with crude haemolymph/ gut juices or higher concentration of haemolymph/ gut juices there is more degradation of dsRNA in comparison to control and low concentration of haemolymph. Which means that the dilution of haemolymph/ gut juices is inversely proportional to the level of degradation. Complexation of nanocarrier with dsRNA enhance the stability of the same by encapsulation, tethering and thereby shielding it from nuclease attack in the gut or in other terms reducing the exposure of dsRNA to DSRNA specific nucleases. Secondly as few studies report the endosomal trapping of the dsRNA once it enters the lepidopteran cells and thus it is not processed into siRNA by the RNAi machinery, the nanoparticles may been playing a key role in bypassing this endosomal pathway, thereby improving the RNAi efficiency in these insect species. We have elaborated the same in the MS in the discussion part as suggested by the reviewer in addition to already discussed text in context to this comment (Line 377 to Line 390)
Further our previous studies based on in vitro assays showed clear data (dsRNA stability experiments) about dsRNA stability before and after shielding with nanoparticles and these articles have been referred in the respective places in support of our present study. Also, a recent study (Kolge et al. 2021) applied chitosan-shielded dsRNA on the field and showed very good mortality in lepidopteran insects. Some of the relevant literature cited is as follows
References:
- Gurusamy, D.; Mogilicherla, K.; Palli, S.R. Chitosan nanoparticles help double-stranded RNA escape from endosomes and improve RNA interference in the fall armyworm, Spodoptera frugiperda. Arch. Insect Biochem. Physiol.2020, 104, e21677, doi:10.1002/arch.21677.
- Kaur, R.; Gupta, M.; Singh, S.; Joshi, N.; Sharma, A. Enhancing RNAi Efficiency to Decipher the Functional Response of Potential Genes inBemisiatabaciAsiaII-1 (Gennadius) Through dsRNA Feeding Assays. Front. Physiol. 2020, 11, 123, doi:10.3389/fphys.2020.00123.
- Gurusamy, D., Mogilicherla, K., Shukla, J.N. and Palli, S.R., 2020. Lipids help double‐stranded RNA in endosomal escape and improve RNA interference in the fall armyworm, Spodoptera frugiperda. Archives of Insect Biochemistry and Physiology, 104(4), p.e21678.
- Das, S.; Debnath, N.; Cui, Y.; Unrine, J.; Palli, S.R. Chitosan, Carbon Quantum Dot, and Silica Nanoparticle Mediated dsR-NA Delivery for Gene Silencing in Aedes aegypti: A Comparative Analysis. ACS Appl. Mater. Interfaces, 2015, 7, 19530–19535, doi:10.1021/acsami.5b05232.
- Kolge, H.; Kadam, K.; Galande, S.; Lanjekar, V.; Ghormade, V. New frontiers in pest control: Chitosan nanoparticles-shielded dsRNA as an effective topical RNAi spray for Gram Podborer biocontrol. ACS Applied Bio Materials, 2021. 4(6), 5145-5157.
Reviewer comment 5: Is the pest mortality attributable to RNAi-inhibited gene expression? The author should provide control experiment results to exempt the effect of nanoparticles.
Author’s response: We agree to the reviewer comment that most of the gene knockdown cases may not result in the mortality of pest, unless and until a unique trait governed by the single vital gene is not targeted. As per the reviewers suggestions we have fed all the nanoparticles alone the insect to overrule their impact on the insect (Supplementary table S5 and Supplementary Figure S13
Reviewer comment 6: I recommend the author to add a schematic illustration to highlight the logic, concept and mechanism of this study, so that people can capture the main point quickly, and the work will make a greater impact in the field.
Author's response: Graphical abstract illustrating the complete concept of MS has been included
Reviewer 2 Report
In this manuscript, candidate reference genes were identified and validated across different development stages using various algorithms. It also confirmed that coupling dsRNA with nanoparticles such as chitosan, lipofectamine, and CQD would improve the RNAi efficacy. This would hasten the transition of RNAi technology from the lab to the field. The experimental design was scientifically sound. It can be accepted for publication in minor revision.
1.A lot of abbreviations were used in the context and a list of abbreviations should be given.
2.In Figure 6, the numbers 1, 2, 3, 4, 5, 6 should be explained in the Figure legend.
Author Response
Open Review 2 Comments and Suggestions for Authors
In this manuscript, candidate reference genes were identified and validated across different development stages using various algorithms. It also confirmed that coupling dsRNA with nanoparticles such as chitosan, lipofectamine, and CQD would improve the RNAi efficacy. This would hasten the transition of RNAi technology from the lab to the field. The experimental design was scientifically sound. It can be accepted for publication in minor revision.
Author’s response: We appreciate reviewers for encouraging the significance of the work and article writeup.
Reviewer comment 1: A lot of abbreviations were used in the context and a list of abbreviations should be given.
Author’s response: We have listed the abbreviations in MS
Reviewer comment 2: In Figure 6, the numbers 1, 2, 3, 4, 5, 6 should be explained in the Figure legend.
Author's response: Needful done with in Legend of Figure 6; modified as suggested
Reviewer 3 Report
Overall presented well. Mostly minor comments (attached) like: [1] reporting gene and protein names appropriately, [2] figures and tables cited correspond to correct ones in text or supplemental, [3] lots of grammatical errors to correct, like proper scientific names and words running together.
Also, had one question about feeding. Was the amount of ingested food quantified after the organisms were fed for 5 days? It is possible that some nanoparticles worked better than others simply because they were more palatable and the insects preferred one over the others. Also, report which nanoparticle type worked the best if there was a difference and if so, why?
Methodologies and descriptions are typical for this type of study.

Author Response
Open Review 3 Comments and Suggestions for Authors
Overall presented well. Mostly minor comments (attached) like:
Author’s response: Thanks to reviewers for encouraging the significance of the work and article write-up.
Reviewer comment 1: reporting gene and protein names appropriately.
Author’s response: According to reviewer suggestions, we cross-checked the gene names and protein names in the entire text.
Reviewer comment 2: figures and tables cited correspond to correct ones in text or supplemental.
Author's response: The figures and table's corresponding citations in the entire text have been thoroughly cross-checked.
Reviewer comment 3: lots of grammatical errors to correct, like proper scientific names and words running together.
Author's response: The entire MS has been checked for grammatical and language errors as well as scientific names
Reviewer comment 4: Also, had one question about feeding. Was the amount of ingested food quantified after the organisms were fed for 5 days? It is possible that some nanoparticles worked better than others simply because they were more palatable and the insects preferred one over the others. Also, report which nanoparticle type worked the best if there was a difference and if so, why? Methodologies and descriptions are typical for this type of study.
Author's response: We didn’t observed any difference in the palatability of the nanoparticles as all the insects irrespective of the nanoparticle used consumed an average of 306± 6.2 to 316±3.8 mg semisynthetic diet in 5 days. The efficacy of the nanoparticles in terms of the gene knockdown has been discussed in detail. In terms of mortality and phenotypic changes it was apparent only in the chitosan nanoparticles conjugated with dsRNA against chitinase gene and this has been highlighted in the text Line 456 to Line 461.
Reviewer 4 Report
The authors present data on the transcriptome and subsequent generation of dsRNAs against various targets in Earias vittella. This data in itself is of high value in providing transcriptomic data of yet another lepidopteran species. The authors then proceed to compare these dsRNA targets with and without the addition of various nanoparticle formulations and measure down regulation. This is an important topic as being able to increase the toxicity of dsRNA to lepidoptera would provide another welcomed tool for lepidoptera control. The MS is relatively well written, but there are misspellings and many of the genus and species names run together.
The primary concern with this manuscript by this reviewer is the lack of toxicity/mortality data for all targets with/without nanoparticles (lines 225-226). Without this data it will be difficult to determine whether any of these dsRNA's with nanoparticles could "..efficiently manage this pest" as the authors state on lines 30-31 in the abstract. The authors state that the only "naked" dsRNA with toxicity was for chitin synthase (Fig. 7) but it would have been nice to have a table showing lack of activity of the other targets (So the authors should have this data). But if the main theme of this MS is to demonstrate the value of the addition of nanoparticles, providing data on the mortality of these dsRNA nanoparticles is critical to determine which dsRNA target is most toxic (and compare these results to published results) and which should be possibly pursued for field applications. As the authors generated mortality data for naked dsRNA's, it is surprising to this reviewer that mortality with dsRNA nanoparticles was not conducted (or reported).
Specific comments:
Lines 53-57: These species are not orders. Please include the orders of these species
Line 470: please provide a reference or list of ingredients of the artificial diet
line 535: The authors state they were going to determine the "most efficient gene" based on expression. Wouldn't the toxicity of the specific dsRNA also figure into "most efficient"? Please revise or provide rationale for this statement.
Author Response
Open Review 4 Comments and Suggestions for Authors
The authors present data on the transcriptome and subsequent generation of dsRNAs against various targets in Earias vittella. This data in itself is of high value in providing transcriptomic data of yet another lepidopteran species. The authors then proceed to compare these dsRNA targets with and without the addition of various nanoparticle formulations and measure down regulation. This is an important topic as being able to increase the toxicity of dsRNA to lepidoptera would provide another welcomed tool for lepidoptera control. The MS is relatively well written, but there are misspellings and many of the genus and species names run together.
Author’s response: Thanks to reviewers for encouraging comments. The spellings and genus and species names have been thoroughly checked and corrected throughout the MS
Reviewer comment: The primary concern with this manuscript by this reviewer is the lack of toxicity/mortality data for all targets with/without nanoparticles (lines 225-226). Without this data it will be difficult to determine whether any of these dsRNA’s with nanoparticles could “efficiently manage this pest” as the authors state on lines 30-31 in the abstract.
Author's response: It has been well established in lepidopteran insects are poor responsive to RNAi and many studies as well as our have stated that the degradation of dsRNA by the gut nucleases is the first hurdle in this process followed by endosomal trapping which leads to the degradation of dsRNA in the insect gut. The mortality or the phenotypic effect depend on the vitality of the selected gene and in our case the phenotypic effects were apparent only with dsChitin synthase conjugated with Chitosan. We are in process of screening more targets for our future studies for managing this pest in the field. However we have added the data for various genes in combination with various nanoparticles (Supplementary table S5) as well as we have also conducted studies to overrule any impact of nanoparticle on the insect (Supplementary Figure S13 and Table S5
Reviewer comment: The authors state that the only “naked” dsRNA with toxicity was for chitin synthase (Fig. 7) but it would have been nice to have a table showing lack of activity of the other targets (So the authors should have this data). But if the main theme of this MS is to demonstrate the value of the addition of nanoparticles, providing data on the mortality of these dsRNA nanoparticles is critical to determine which dsRNA target is most toxic (and compare these results to published results) and which should be possibly pursued for field applications. As the authors generated mortality data for naked dsRNAs, it is surprising to this reviewer that mortality with dsRNA nanoparticles was not conducted (or reported).
Author's response: We appreciate the reviewer for raising a valid concern. We have conducted bioassays with all the genes wherever we have studied gene knockdown, however unfortunately we could not observe mortality/ phenotypic changes with these genes except for dsChitnase-Chitosan complex. As suggested we have included that data in tabular form for each gene as well as the nano-particles alone to overrule any effect of these nanoparticles on studied insect (Supplementary figure S13 and Table S5).
Specific comments:
- Lines 53-57: These species are not orders. Please include the orders of these species.
Author’s response:. According to the reviewer's suggestion, we have done the needful.
- Reviewer comment: Line 470: please provide a reference or list of ingredients of the artificial diet.
Author's response: Needful done
Reference:
Gupta, G.P.; Rani, S.; Birah, A.; Raghuraman, M. Mass rearing of the spotted bollworm, Earias vittella (Lepidoptera: Noctuidae) on an artificial diet. Int. J. Trop. Insect Sci. 2005, 25, 134-137.
- line 535: The authors state they were going to determine the “most efficient gene” based on expression. Wouldn’t the toxicity of the specific dsRNA also figure into most efficient? Please revise or provide rationale for this statement.
Author's response: The “most efficient gene” related to suitable reference for using the normalization of the qRT-PCR candidate genes expression data. It won't show any toxicity to specific dsRNA. According to reviewer suggestions, we changed the sentence “The most efficient reference gene was selected from the candidate reference genes based on their expression stability.”
Round 2
Reviewer 4 Report
In this resubmitted manuscript, the authors have positively addressed the concerns of this reviewer which is much appreciated. Several minor comments on Tables and Figures
Table 1: Please describe in the caption what the numbers mean in the table
Fig. 6: In the first sentence of the caption, please include a statement that states something about comparisons with and without nano-particle coating which is the primary goal of this Figure. Also, please state what KD means.
Fig. 7. In the first sentence of the caption, please include a statement that states something about comparisons with and without nano-particle coating which is the primary goal of this Figure.
Author Response
Reviewer Comment: Table 1: Please describe in the caption what the numbers mean in the table
Author's response: Needful done, Footnote given at the end of table 1
Reviewer Comment: Fig. 6: In the first sentence of the caption, please include a statement that states something about comparisons with and without nano-particle coating which is the primary goal of this Figure. Also, please state what KD means.
Author's response: Needful done as suggested
Reviewer Comment: Fig. 7. In the first sentence of the caption, please include a statement that states something about comparisons with and without nano-particle coating which is the primary goal of this Figure.
Author's response: Revised as suggested